# COVID-19 and its impact on the national examination for pharmacists in Japan: An SNS text analysis

Tomoya Kitayama *

School of Pharmacy and Pharmaceutical Sciences, Mukogawa Women's University, Nishinomiya, Hyogo, Japan

* tomokita@mukogawa-u.ac.jp

## Abstract

The COVID-19 pandemic has created an extraordinary situation for undergraduate students. The aim of this study is to evaluate the impact of the COVID-19 pandemic on the national examination for pharmacists in Japan. In this study, we analyzed the content of Twitter to assess the impact of COVID-19 on the national exam, including psychological aspects. Tweets including the words "national examinations" and "pharmacists" were compiled from December 2020 to March 2021. ML-Ask, a python library, was used to evaluate the emotional register of the tweets on the basis of ten elements: Joy, Fondness, Relief, Gloom, Dislike, Anger, Fear, Shame, Excitement, and Surprise. The presence of COVID-19-related terms was clearly visible in tweets about the national examination of pharmacists between December 1st–and 15th, 2020. It was precisely during this period that the government had announced a strategy regarding national examinations, in the light of COVID-19. The analysis found that post December 16th, words associated with negative emotions were mainly related to the examination, but not to COVID-19. As a result of analyzing only infected areas, a relationship between employment and negative feeling was detected.

## Introduction

Since 2000, there have been several coronavirus epidemics in the world. For example, there was the Severe Acute Respiratory Syndrome Coronavirus (SARS-CoV) from 2002 to 2003, H1N1 influenza in 2009, and the Middle East Respiratory Syndrome Coronavirus (MERS-CoV) in 2012. The epidemics of these viruses, unlike COVID-19, did not become a pandemic in Japan. COVID-19 was a "novel Coronavirus", and the first case of COVID-19 was reported in December 2019 from Wuhan City in China [1]. The COVID-19 pandemic went on to affect the entire world, and Japan was no exception.

The pandemic not only impacted healthcare workers, but also had a major impact on undergraduate education. Regarding pedagogy, there has been a transition to remote, online education, and its impact has been analyzed. Further development of new educational programs is underway [2–5]. Healthcare undergraduate students are required to take and pass a national examination. However, no detailed study has been done on the impact of COVID-19 on national examinations. In the three years from 2018, the average number of applicants for

**Data Availability Statement:** Please obtain the data set (Remote lessons; Face-to-face lessons; National exam; prefecture) provided for the analysis from public repository (https://github.com/tomokitamukogawa/twitterdata). The data set

provided is the data necessary for analysis, excluding the parts connected to individuals such as user names and IDs.

**Funding:** The author(s) received no specific funding for this work.

**Competing interests:** The authors have declared that no competing interests exist.

the national examination for doctors was 9,333. The average number of applicants for the national examination for pharmacists was 15,486, approximately 1.5 times the number of applicants for the national examination for doctors [6]. Comparing the pass rates of these exams, compared to 2018 before COVID-19, in 2020 there was a slight increase in applications for the national examination for doctor (90.1% → 92.1%) and a decrease in applications for the national examination for pharmacists (70.58% → 69.58%). The number of pharmacists (321,982 as of 2020) is almost the same as the number of doctors (339,623 as of 2020), and they are important healthcare workers in Japan. To analyze the impact of COVID-19 on national examinations, this study focused on the national pharmacist examination viewed from the factors of the number of applicants and the pass rate.

The purpose of this study is to analyze the impact of COVID-19 on national examinations, including psychological aspects, on a nationwide scale in Japan. It is important to study what kind of impact the global pandemic and accompanying social environmental changes have had on future pharmacists, so as to make sure things are improved in an eventual next time. Several studies reported the psychological stress caused by the pandemic in pharmacy students [7–9]. Japanese pharmacy students also feel stress, and the purpose of this study is to clarify the psychological conditions in various fields by analyzing the national examinations. This may include the suspension and/or rescheduling of planned final year class, adapting to online virtual learning, anxiety about taking national examinations, anxiety about infection, and expectations and worries about becoming a pharmacist. Previous studies analyzing the impact of COVID-19 on education have often used questionnaires. However, limitations of these studies include questionnaire content, scope of data collection, and choice of data collection subjects. In this study, text was automatically collected from Twitter and analyzed. This is because analyzing the collected text can reveal a broader range of emotional changes than using multiple choices.

## Methods

### Study sample size

The tweets to be analyzed were collected for four months between December 1st, 2020 and March 31st, 2021. The national examination for pharmacists was held on February 20th and 21st, and the results were announced on March 24th, 2021. Tweet data was collected by Python 3.8.5 (https://www.python.org/). The Python-powered libraries used were tweepy, schedule, pytz and pandas. Using Python, the data was generated as a.csv file (S1 Appendix). It is important to note that not all relevant tweets were collected due to Twitter API limitations, internet environment issues, etc. Collecting tweets was done by the Python program, regardless of the intention of the author. Similarly, tweets that failed to be collected were mechanical problems and occurred randomly regardless of the author's intention.

The study database collected 27,494 tweets that included the terms "remote lessons" and "university", and 16,561 tweets containing "face-to-face lessons" and "university", to investigate the impact on university students in general. Additionally, a total of 7,326 tweets contained the terms "national examinations" and "pharmacists". Areas where a state of emergency was declared (based on the content described in the tweet_user_description) were identified from the 7,326 tweets containing "national exam" and "pharmacist". Here, only tweets in which city names or prefecture names were recognized in the tweet_user_description (Table 1) were included.

### Analysis of tweets

The collected tweets were analyzed using KH coder (https://github.com/ko-ichi-h/khcoder) [10–12] and Python 3.8.5. The Python-powered library used was ML-Ask (https://github.com/ikegami-yukino/pymlask) [13]. KH Coder is software for quantitative text analysis or text

**Table 1. Number of tweets in each prefecture.**

| Prefecture | Tweets | Prefecture | Tweets | Prefecture | Tweets |
|---|---|---|---|---|---|
| Aichi* | 116 | Kagawa | 19 | Osaka* | 178 |
| Akita | 22 | Kagoshima | 13 | Saga | 8 |
| Aomori | 110 | Kanagawa* | 344 | Saitama* | 484 |
| Chiba* | 111 | Kochi | 24 | Shiga | 36 |
| Ehime | 77 | Kumamoto | 79 | Shimane | 4 |
| Fukui | 27 | Kyoto* | 46 | Shizuoka | 47 |
| Fukuoka* | 85 | Mie | 29 | Tochigi* | 47 |
| Fukushima | 27 | Miyagi | 51 | Tokushima | 46 |
| Gifu* | 45 | Miyazaki | 15 | Tokyo* | 554 |
| Gunma | 37 | Nagano | 41 | Tottori | 5 |
| Hiroshima | 43 | Nagasaki | 18 | Toyama | 102 |
| Hokkaido | 108 | Nara | 23 | Wakayama | 72 |
| Hyogo* | 147 | Niigata | 41 | Yamagata | 26 |
| Ibaraki | 41 | Oita | 32 | Yamaguchi | 68 |
| Ishikawa | 38 | Okayama | 37 | Yamanashi | 16 |
| Iwate | 36 | Okinawa | 14 | **unclear** | 3,737 |
| | | | | **Total** | **7,326** |

*: Areas with a high rate of infection where a state of emergency was declared during the research period.

mining, and was used for cross-tabulation, correspondence and a co-occurrence network. KH Coder is written in Perl and uses ChaSen, MeCab, TermExtract, MySQL, R language, etc., as backends. ChaSen, MeCab and TermExtract are morphological analysis engines, MySQL is a relational database management system, and R language is a programming language for statistics. ML-Ask, a python library, was used for sentiment analysis of tweets. ML-Ask performs morphological analysis of the input text and compares it with the emotion dictionary database, making it possible to classify the text into 10 axes: "Joy", "Fondness", "Relief", "Gloom", "Dislike", "Anger", "Fear", "Shame", "Excitement", and "Surprise". Based on Contextual Valence Shifters, it performs contextual emotion estimation. For example, in the case of the sentence "I can't say I like it", since like is negated, ML-Ask infers that it is dislike, which is the opposite feeling of like [13]. Data from the analysis results were generated as .csv files using Python (S1 Appendix). Each tweet contained symbols and tag information that has no meaning in Japanese. Therefore, the data was preprocessed to remove symbols and tag information that would cause errors in the analysis. In the trial analysis by KH coder, some words were detected with inappropriate delimiters, so forced extraction words were set (S2 Appendix).

The categories used in the cross-tabulation were selected from co-occurrence networks of all related words collected. For certain nouns, paraphrases were also included in the category (Covid, COVID-19, etc.). The categories used in the cross-tabulation were:

1. Coronavirus-related: Corona, Covid, COVID-19, delta, infect, infection spread, infected person, onset, positive, cluster, close contact, fever, isolation, recuperation, spread, emergency, and state of emergency

2. Relief measures-related: additional test, reexamination, another day, and relief measures

3. Exam-related: exam, examination, difficulty, evaluation, score, grading, and self-scoring

4. Friendship-related: friend, close friend and classmate.

In the given time period, the number of times each of the above words occurred was counted. Differences in ratios for each period were tested for significance by residual analysis. The association coefficient (Cramer's V) was calculated as the effect size [14]. The residual analysis was performed by the following formulas:

$$Expected\ value\ (E_{ij}) = \left(\sum_{i=1}^{a} n_{ij} \times \sum_{j=1}^{b} n_{ij}\right) / \sum_{i=1}^{a} \sum_{j=1}^{b} n_{ij}$$

$$Residuals\ variance\ \left(R_{ij}\right) = \left(1 - \frac{\sum_{i=1}^{a} n_{ij}}{\sum_{i=1}^{a} \sum_{j=1}^{b} n_{ij}}\right) \times \left(1 - \frac{\sum_{j=1}^{b} n_{ij}}{\sum_{i=1}^{a} \sum_{j=1}^{b} n_{ij}}\right)$$

$$x_{ij} = (n_{ij} - E_{ij}) / \sqrt{E_{ij} \times R_{ij}}$$

$$p\ value = 2 \times \left(1 - \left(\frac{1}{\sqrt{2\pi}} e^{-\frac{x_{ij}^2}{2}}\right)\right)$$

Cramer's V was calculated by the following formulas:

$$Chi\ square\ value\ (X^2) = \sum_{i=1}^{a} \sum_{j=1}^{b} \frac{(n_{ij} - E_{ij})^2}{E_{ij}}$$

$$Cramer's\ V = \sqrt{X^2 / \left(\sum_{i=1}^{a} \sum_{j=1}^{b} E_{ij} \times (number\ of\ categories - 1)\right)}$$

$n_{ij}$ indicates the value of each cell. (i;columns, a is the maximum value. j;rows, b is the maximum value.)

For the number of categories, we used three in the vertical direction, since the lower number should be used in the calculations.

Emotional recognition of texts by ML-Ask was evaluated on the basis of ten emotional elements. Correspondence analysis was performed using emotional elements as external variables to extract words that are highly relevant to each emotion. This study also included a correspondence analysis using the software KH coder, thus obtaining a more objective classification without being biased by the researchers' perspectives. Correspondence analysis is an analytical method that visualizes a cross-tabulation table, and displays the relationship between the number of times a word appears and a group (emotional element) as a distance. The distance is calculated by dividing the Euclidean distance by the square root of the ratio to the total number of words appearing in each group. Therefore, the whole profile was placed at the origin (0,0), and characteristics were judged by the direction in which the word appeared from that point.

## Ethical considerations

Research using social networking service (SNS) data has increased in recent years, but uniform research ethics guidelines around it are yet to be developed [15]. According to Twitter's terms of service, when users register, they are required to allow third parties to use the content they post. In fact, since the first post was published when the company was founded in 2006, all public posts are searchable on the Internet. In addition, Twitter revised its developer policy in 2020, clearly stating that it is possible to collect posted content for non-commercial research purposes, and clarifying the rules for using academic data. Therefore, in the case of big data analysis, the data is public information, and it is often argued that ethical review is unnecessary.

In recent years, ethical guidelines for research using SNS data have been proposed [16, 17]. Most notably, George Washington University has published research ethics guidelines for using SNS data (George Washington University Libraries, online). The contents of the guidelines are as follows: the rules of the SNS platform must be followed, collection should be limited to public data, consent should be obtained when using data such as direct messages, collection of metadata such as profile information should be kept to a minimum, consent is required when quoting and posting the text of SNS, and user IDs and account names may only be provided when consent has been obtained. In this study, data was managed according to these guidelines. Therefore, the SNS text, user ID, and other identifying data is not described in the text, nor is it provided to third parties.

## Limitations of this study

A limitation of this study was the small number of accounts (3471) in the national examination for pharmacists tweet analysis. This number was approximately 22.4% of applicants for the national examination for pharmacists, and included accounts whose applicants could not be determined. In the analysis comparing areas where the state of emergency was declared and other areas, tweets (51%) whose prefecture could not be identified were excluded from the analysis.

## Results and discussion

### Impact of COVID-19 in Japan during the study period

As of March 31st, 2021, Japan reported a total of 472,947 Covid-19 infections [18]. A "third" outbreak of infectious disease occurred during the study period (Fig 1). A new variant was

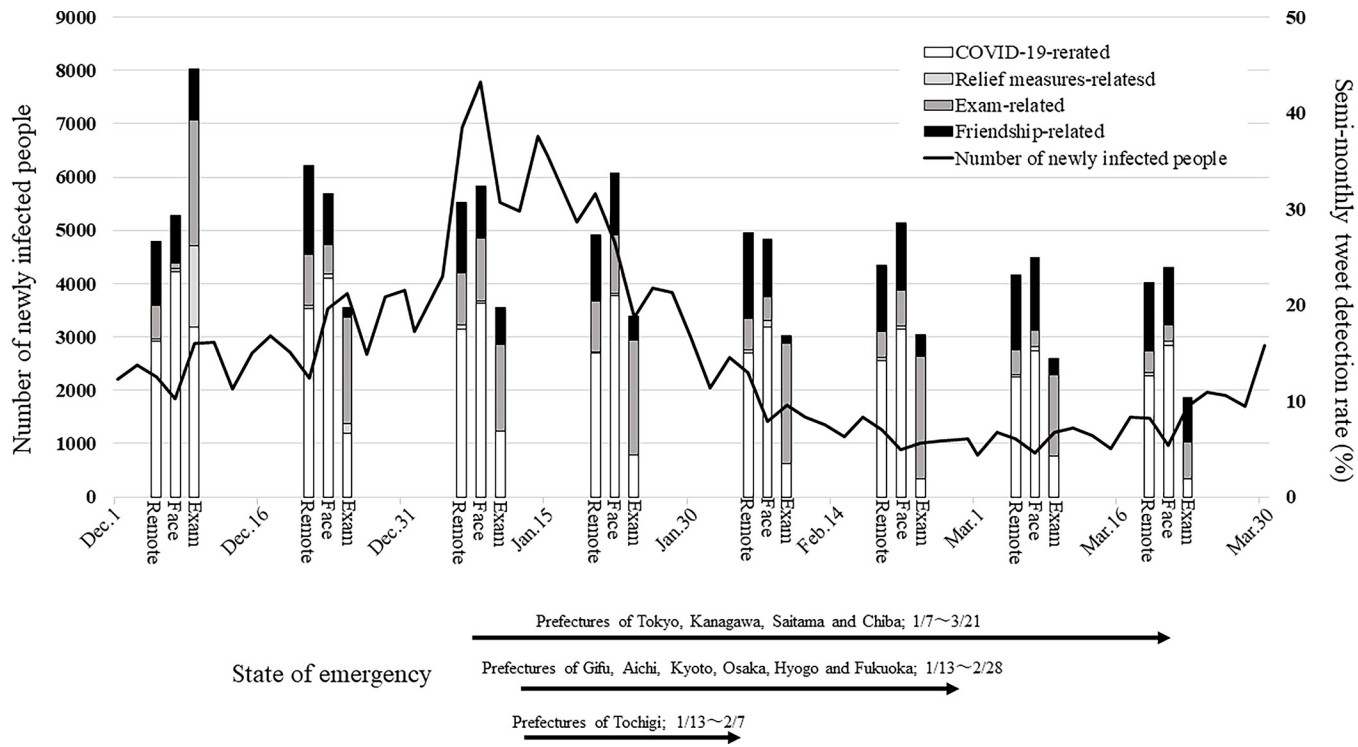

**Fig 1. Number of newly infected people and impact on the national examination for pharmacists.**

detected at an airport on 25 December, and an outbreak was confirmed in an urban area on 30 January [19]. In early December, a shortage of medical workers caused a medical crisis, and several prefectures applied for the dispatch of medical workers to the Japanese Self-Defense Forces. On January 23, the cumulative number of deaths due to COVID-19 exceeded 5,000. The government issued a state of emergency declaration in 11 of the 47 prefectures. A state of emergency was declared in Tokyo, Kanagawa, Saitama, and Chiba from January 7th, 2021 to March 21th, 2021. Similarly, declarations were made in other areas at different times.

Regarding vaccines important for infection prevention, an application for approval of Pfizer's vaccine was submitted to the Ministry of Health, Labor and Welfare on December 18. From February 17th, vaccination was implemented only for medical workers. On the other hand, vaccinations at universities, including students taking the national examination for pharmacists, started on June 21, three months after the examination was conducted.

In November, the Ministry of Education, Culture, Sports, Science and Technology issued a notice to take remedial measures, such as conducting make-up exams, for university entrance exams, one of the major exams to be conducted during the study period. In early December 2020, the Minister of Health, Labor and Welfare announced that the government would not implement an additional examination or other relief measures for the national examination for pharmacists to be held on February 20th and 21st for people infected with COVID-19. Therefore, applicants for the national examination for pharmacist took the examination without out any relief measures while unvaccinated amid medical shortages.

A line graph shows the number of newly infected people. The column graph shows the percentage of tweets containing the object-related phrase in the half-month tweets. The respective numerical values are given in the table in S3 Appendix. The open column shows COVID-19-related tweet percentage, the gray column shows relief measures-related tweet percentage, the dark gray column shows exam-related tweet percentage, and the black column shows friendship-related tweet percentage. The meaning of Remote is a study tweets database that contains the terms "remote lessons" and "university", Face is a study tweets database that contains "face-to-face lessons" and "university", and Exam is a study tweets database that contains "national examinations" and "pharmacists".

## Impact on the national examination for pharmacists

The number of people infected with COVID-19 increased during the first half of December 2020 and peaked in the first half of January (Fig 1). In response to this, tweets related to remote and face-to-face classes maintained a high percentage of new coronavirus-related words, but the decrease in the number of infected people also reduced the appearance rate of new coronavirus-related words (Fig 1, S3 Appendix). Among the tweets about the national pharmacists examination, the number of COVID-19 related words was significantly higher from December 1 to 15. This increase had been declining since late December, despite an increase in the number of infected people. Between December 1st and 15th there was a marked increase in words relating to relief measures in tweets about the national examination for pharmacists. Words associated with relief measures did not increase significantly in all collected tweets during the other time periods. This suggested that coronavirus-related and relief measures-related changes in the national examination are more affected by social conditions than by changes in the number of newly infected people. During this period, the Minister of Health, Labor and Welfare announced that in relation to the national examination for pharmacists, no relief measures such as additional examinations and alternative examination dates would be implemented for infected persons. It is highly probable that this announcement had a great impact on the examinees. For exam-related words, the peak was observed in the first half of December

2020. This suggested that the peak was related to the announcement by the Minister of Health, Labor and Welfare, but was not at a significant level. It is also speculated that the general level of interest in the national examination was high throughout the research period. Finally, changes in the number of words which were friendship-related peaked at different periods for each group of tweets.

## Analyzing the emotional impact of the national examination for pharmacists on examinees

From December 1st to 15th, there were many words related to COVID-19 (Fig 1). "COVID-19" and "infection" were associated with fear and surprise (Fig 2A). "Every day" and "management" related to daily health care and temperature recording were similarly associated with feelings of fear and surprise. The word "exam" itself was found to be weakly associated with fear, but examination-related words were associated with various emotions such as shame, anger, gloom, and fondness. "Family" and "nice project" are related to their dislike feelings, and it was inferred that tweets containing these were in a negative context due to the influence of COVID-19. From December 16th to 31st, there was a notable drop in tweets related to COVID-19 and relief measures. As a result, no words strongly associated with fear were detected (Fig 2B). Feelings to dislike changed from words related to personal life to those related to the examination. Analysis of the tweets in January showed that there was an association between negative emotions and words related to the examination (Fig 3). COVID-19 infections peaked in January 2021, however COVID-19-related words such as "COVID-19" and "infection" were not found to be associated with any emotion since December 16. This data indicates that examinees might have been more influenced by information about the national examination than by the increase in new infections.

In tweets from February 1 to 15, phrases related to the difficulty of the exam, such as "question range" and "easy", were identified with the emotion of anger (Fig 4A). Also, "hard" and

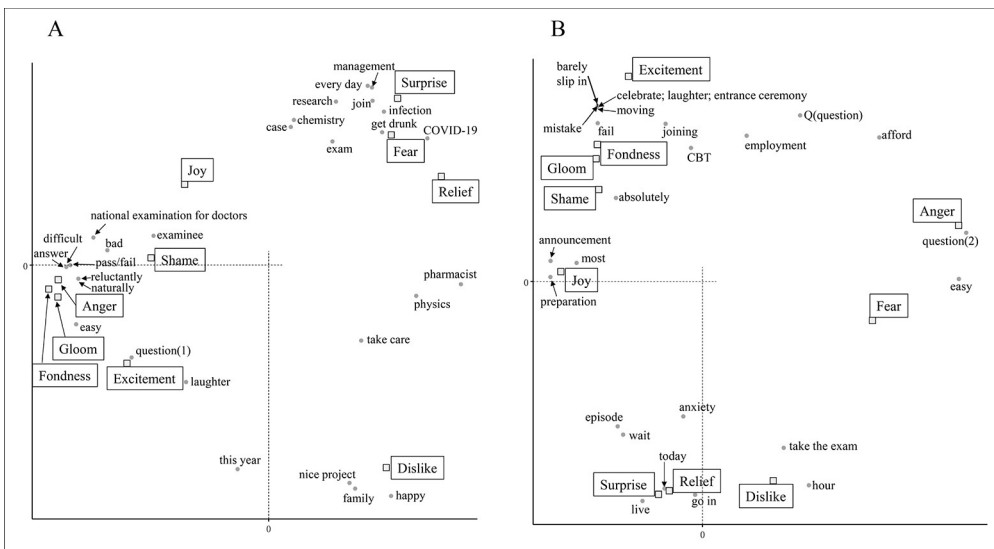

**Fig 2. Analysis of the emotional impact of the national examination for pharmacists on examinees in December 2020.** Correspondence analysis shows the relationship between the examinee's emotions and the words contained in the tweet. Each figure indicates the results for the following periods: A: 1 to 15 December and B: 16 to 31 December. The annotations for conversion from Japanese are shown below. The meaning of question (1) is a sentence that is used to examination, and question (2) is a phrase used when asking someone a question.

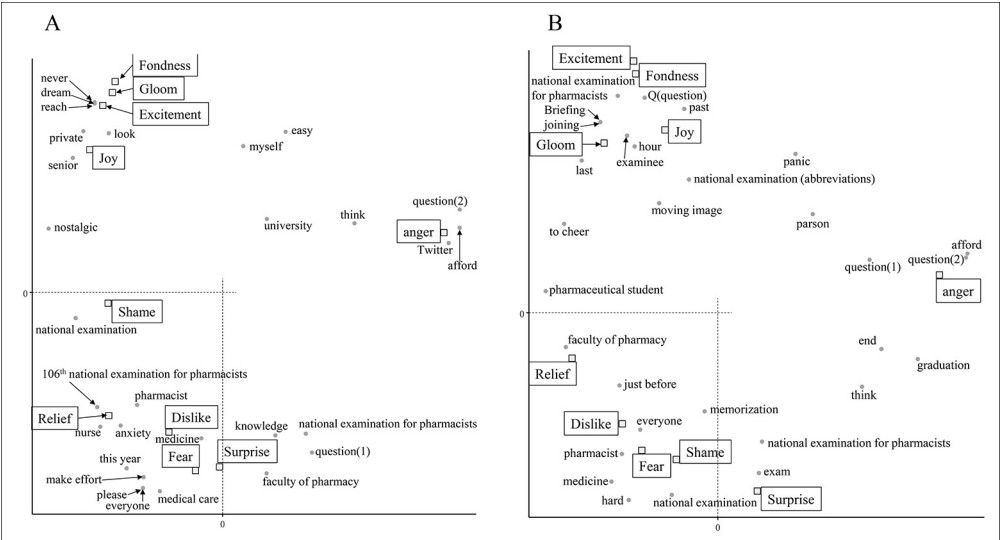

**Fig 3. Analysis of the emotional impact of the national examination for pharmacists on examinees in January 2021.** Correspondence analysis shows the relationship between the examinee's emotions and the words contained in the tweet. Each figure indicates the results for the following periods: A: 1 to 15 January and B:16 to 31 January. The annotations for conversion from Japanese are shown below. The meaning of question (1) is a sentence that is used to examination, and question (2) is a phrase used when asking someone a question.

"examinee" were related to their fear feelings. Given that this period was right before the national examination, tweets were also detected that people expressed positive feelings that they solved the questions. After February 16th, "hard" and "study" were also detected in association with the emotion of fear (Fig 4B). Words detected in association with feelings of anger were related to mental states such as "anxiety" and "afford". It is suggested that this expresses

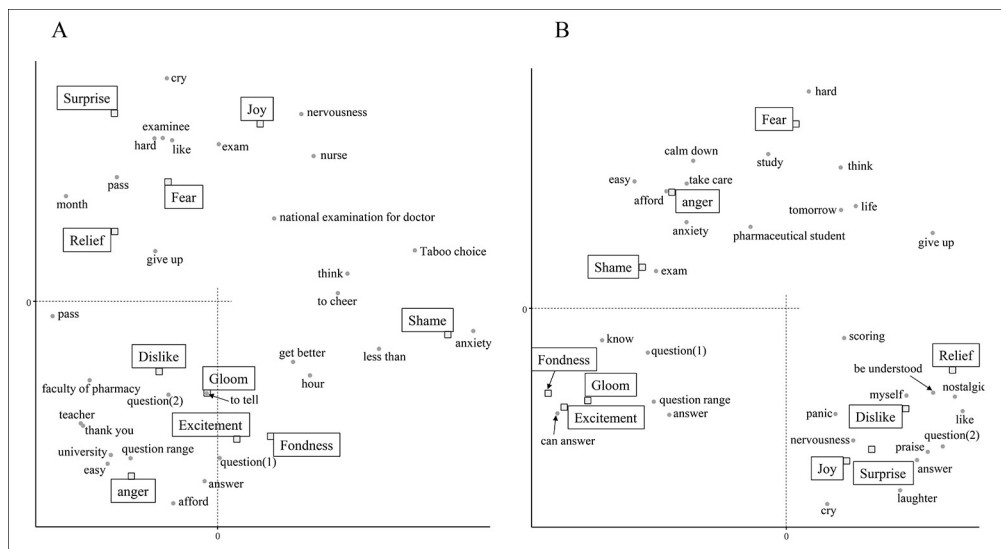

**Fig 4. Analysis of the emotional impact of the national examination for pharmacists on examinees in February 2021.** Correspondence analysis shows the relationship between the examinee's emotions and the words contained in the tweet. Each figure indicates the results for the following periods: A: 1 to 15 February, B:16 to 28 February. The annotations for conversion from Japanese are shown below. The meaning of question (1) is a sentence that is used to examination, and question (2) is a phrase used when asking someone a question.

feelings about one's own impatience for the national examination. Similarly, phrases related to impatience such as "panic" and "nervousness" were detected as emotions of joy and surprise. It is speculated that many tweets were posted in the context of denying "panic" and "nervousness". On the other hand, it was observed that the examinees felt excitement and surprise when they were able to "answer". These results suggested that they had various feelings about taking the exam itself.

The period from March 1st to 15th is the period of waiting for announcement of the results after finishing the self-assessment. "Study" and "worries" were detected in relation to negative emotions such as fear (Fig 5A). It was speculated that this was the emotion associated with the result of self-grading. "Stop" was detected in association with joy and relief, but this word was observed beginning in March. Since "study" was associated with negative emotions, "stop" was probably less relevant to "able to stop studying." The examinees may have been happy that they were able to "stop" what they had been doing other than studying before the national examination. Many words were associated with the feeling of shame. The associated words were "enterprise", "company", "finding employment", "way", "start", etc., and there were many words related to going out from university into life to society. In addition, "anxiety" was also associated with feelings of shame, suggesting that these words were related to "anxiety". After March 16, a similar tendency was observed for "stop", which was associated with feelings of joy and relief (Fig 5B). However, only "anxiety" was detected to have a strong association with feelings of shame. At the announcement of results in late March, it was observed whether the examinee passed or failed, and the various emotions that they produced.

## Analyzing the emotional impact on examinees in high infection areas

Both state of emergency areas and non-state of emergency areas were analyzed throughout the period of the study (Fig 6). The main difference between state of emergency areas and other areas was the words associated with positive emotions of joy and relief, and the words

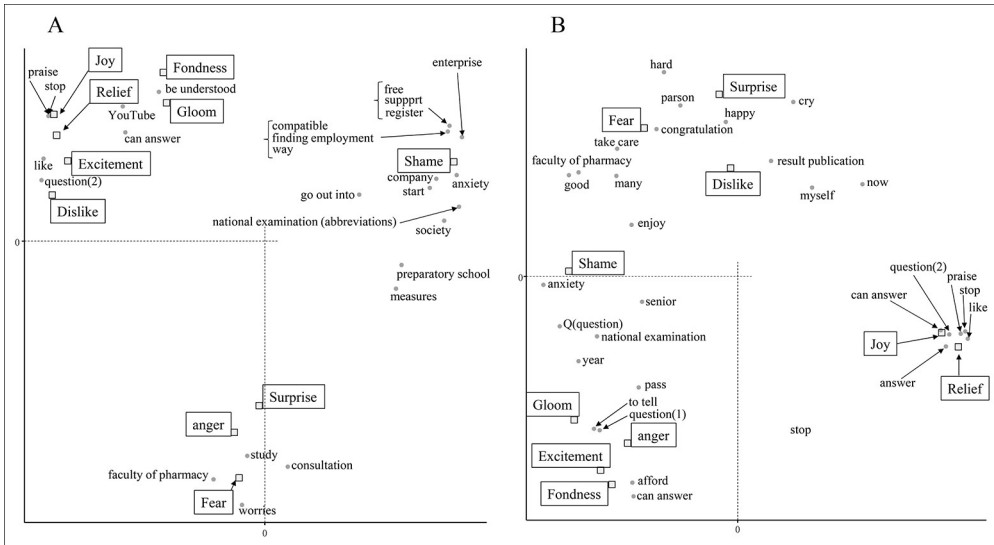

**Fig 5. Analysis of the emotional impact of the national examination for pharmacists on examinees in March 2021.** Correspondence analysis shows the relationship between the examinee's emotions and the words contained in the tweet. Each figure indicates the results for the following periods: A: 1 to 15 March, B:16 to 31 March. The annotations for conversion from Japanese are shown below. The meaning of question (1) is a sentence that is used to examination, and question (2) is a phrase used when asking someone a question.

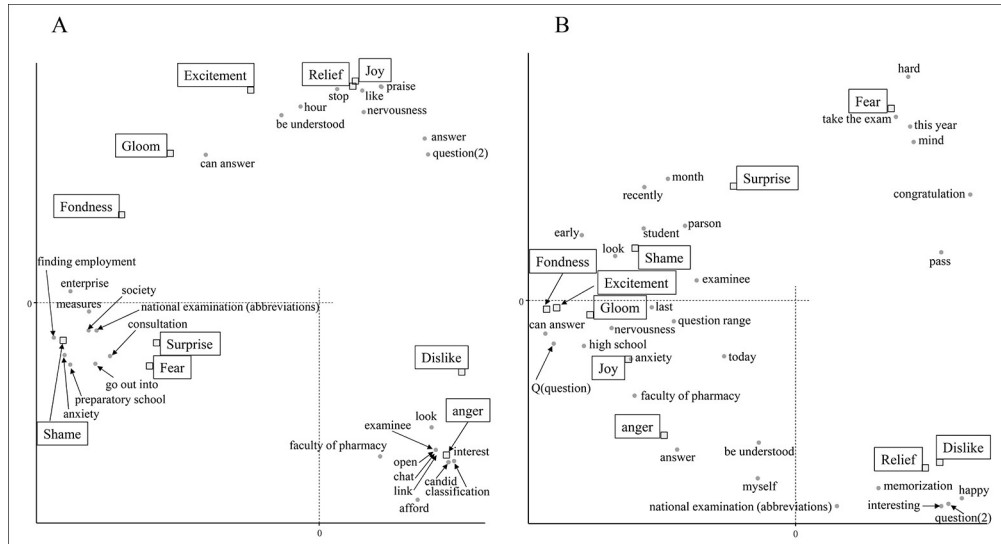

**Fig 6. Analyzing the emotional impact on examinees in high infection areas.** Correspondence analysis compares the relationship between the examinee's emotions and the words contained in the tweets and identifies similarities and differences between high-infection areas and other areas. A) High infection areas were where a state of emergency was declared. B) Areas where no state of emergency was declared. Annotations for conversion from Japanese are shown below. The meaning of question (1) is a sentence that is used to examination, and question (2) is a phrase used when asking someone a question.

associated with negative emotions of shame and fear. In the state of emergency areas, words such as "stop" and "nervousness" came up in association with positive feeling (Fig 6A). In the other areas, it was shown that "memorization" and "question" were related to feelings of relief, and that "anxiety" was strongly associated with feelings of joy. The word "nervousness" was detected between gloom and joy, and "stop" was unrelated to any emotion and was not detected (Fig 6B). "Stop" was detected in association with emotion in March after the end of the national examination (Fig 5). Further "stop" was detected in March after the end of the national examination in areas where the infection was spreading, so it is possible that they felt relief and joy in stopping the infection control behavior that they had been implementing until the examination. In areas where a state of emergency was declared, words such as "anxiety", "counseling", "employment", and "society" were associated with negative emotions (Fig 6A). "Take the exam" was detected as the word most strongly associated with fear, and "student" and "look" were associated with shame in other areas (Fig 6B). In areas with relatively low infection risk, words related to national exams were associated with negative emotions, but in areas where there was a state of emergency, words related to society were detected. Similarly, internet-related words such as "open", "chat", and "link" were associated with anger in areas where the state of emergency was declared, whereas only "answer" was associated with anger in other areas. It can be inferred that examinees in high infection areas where a state of emergency was declared were anxious not only about the examination, but also about their connection with society.

## Conclusions

This paper analyzed the impact of COVID-19 on examinees of the national pharmacist examination in Japan. Examinees were found to be very interested in the contents of the examination system associated with the spread of COVID-19 infection. This effect was seen strongly in

December 2020, when the exam rules were announced, after which there was a focus on the examination. In the first half of December, COIVD-19-related words were detected in association with fear emotions, but since then mainly exam-related words have been associated with negative emotions. On the other hand, examinees in high infection areas that had a state of emergency were found to be anxious about their connection with society, including "employment". This trend was less pronounced in other areas. The results of this study indicate that due to the impact of the COVID-19 pandemic, attention should be paid to social relevance, including the future of examinees, rather than the examination itself.

Examinees experienced their final year with a curriculum that prioritized infection prevention measures. A study at an American university reported that pharmacy students felt that remote lessons hindered the communication and networking skills they needed for their post-graduation careers [7, 20]. They were very worried about not being able to take advanced classes with their classmates in the final year and the incompleteness of the classes. In addition, it is reported that the news of medical facility during the pandemic has shocked students and some students have negative feelings. Such anxieties are likely to make students feel embarrassed and hesitant about working as medical professionals because of COVID-19. Further, leaving the examinees in hesitation can lead to public health losses.

Several studies have suggested that students' anxiety and hesitation about COVID-19 could be improved with accurate knowledge. Rusgis et al. reported the relationship between the hesitation to receive the corona vaccine and the health information sources [21]. Students who were willing to receive the vaccine utilized scientific journals and school curriculum/coursework, and utilized these sources for COVID-19 information. This report suggests that high information literacy among students may reduce anxiety about pandemics. The Ministry of Internal Affairs and Communications conducted a survey of information sources on COVID-19 among six countries (Japan, America, England, France, Germany, South Korea). As a result, Japan tended to be different from other countries [22]. In Japan, the majority of respondents cited commercial broadcasting as their source of information, while in other countries public broadcasting was the most common. The rate of using specialized organizations such as WHO as sources of information was 8.8% in Japan, while the average value in other countries was 28.4%. It is believed that this background influences the anxiety that Japanese students feel about connecting with society. Teaching correct knowledge, including information literacy, is useful in alleviating students' anxiety [23]. In the future, it will be necessary for pharmacy education in Japan to find ways to address students' hesitation.

## Supporting information

**S1 Appendix. Code used in research.**
(PDF)

**S2 Appendix. Preparation before analysis by KH coder.**
(PDF)

**S3 Appendix. Cross-tabulation between lesson format or national exams and each category.**
(PDF)

## Author Contributions

**Conceptualization:** Tomoya Kitayama.

**Data curation:** Tomoya Kitayama.

**Formal analysis:** Tomoya Kitayama.

**Funding acquisition:** Tomoya Kitayama.

**Investigation:** Tomoya Kitayama.

**Methodology:** Tomoya Kitayama.

**Project administration:** Tomoya Kitayama.

**Resources:** Tomoya Kitayama.

**Supervision:** Tomoya Kitayama.

**Validation:** Tomoya Kitayama.

**Visualization:** Tomoya Kitayama.

**Writing – original draft:** Tomoya Kitayama.

**Writing – review & editing:** Tomoya Kitayama.

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
