## [Decision Letter · Decision Letter 0]

1 Feb 2023

PONE-D-22-28806COVID-19 and its impact on the national examination for pharmacists in Japan: An SNS text analysis

PLOS ONE

Dear Dr. Kitayama,

Thank you for submitting your manuscript to PLOS ONE. After careful consideration, we feel that it has merit but does not fully meet PLOS ONE’s publication criteria as it currently stands. Therefore, we invite you to submit a revised version of the manuscript that addresses the points raised during the review process.

We look forward to receiving your revised manuscript.

Kind regards,

Akira Ehara, Ph.D., M.D.

Academic Editor

PLOS ONE

Journal Requirements:

Additional Editor Comments:

The analysis is based on artificial intelligence, but there is no mention of the analysis method. Presumably, they are investigating the relationship between COVID-19 and emotional expressions in TWITTER's writing to find out which emotion they are close to, but there are no details. It is artificial intelligence, so it is inevitable that it is black boxed to some extent, but please mention its analysis method, even if it is only a few lines.

I can read Japanese, but there is no mention of the analysis method on the website of User Local, Inc written in Japanese (https://emotion-ai.userlocal.jp/) .

Reviewers' comments:

Reviewer's Responses to Questions

**Comments to the Author**

1. Is the manuscript technically sound, and do the data support the conclusions?

Reviewer #1: Yes

Reviewer #2: Yes

Reviewer #3: No

Reviewer #4: Partly

2. Has the statistical analysis been performed appropriately and rigorously? 

Reviewer #1: Yes

Reviewer #2: Yes

Reviewer #3: I Don't Know

Reviewer #4: Yes

3. Have the authors made all data underlying the findings in their manuscript fully available?

Reviewer #1: Yes

Reviewer #2: Yes

Reviewer #3: No

Reviewer #4: Yes

4. Is the manuscript presented in an intelligible fashion and written in standard English?

Reviewer #1: Yes

Reviewer #2: Yes

Reviewer #3: Yes

Reviewer #4: Yes

5. Review Comments to the Author

Reviewer #1: -line 67 Twitter is misspelled

-double check for periods before quotation marks throughout. There are some grammatical errors.

-line 327 needs revision

-good figures to explain fear, happiness, anger, and sadness

Reviewer #2: The study promoted by this work is really interesting. The analysis odf emotional impact connected to the pandemic lapse time is a fruitful line if research from various point of view. The authors have accurately managed the data following a correct procedure. Anyway in my opinion in getting a more robustness of the results much more must be explained for that concerns the AI technical tools which involved in this study. The tweet statement are a right "vector" by which conducts this analysis but in my opinion some Natural Language Processing methods must be exploited and deeply analyzed. Python is a good basis by which wording NLP approach with a deep sentiment analysis of the archived statements on the tweets.

My request is to explain in a detailed way the methods and the related tools were involved in training these precious data in supporting the interesting results. The Journal deserves a rigorous analysis and the topics of this study too.

I am ready in receiving a revised version of the paper for reconsidering my decision on

Reviewer #3: Major Comments

Conclusions and working hypothesis

The paper sets itself to study the effect of COVID-19 on pharmacist examinations in Japan, but makes no effort in explaining why this research question is of general scientific or social relevance. It is clear that this can be of importance for the pharmacist community, but how does this inform us about other examinations in other fields? What is the underlying social or psychological process that this particular case is exemplifying? Furthermore, more information about the scale and nature of the examinations (amount of students, social importance, associated psychological or social phenomena with them, etc.) would make the question of general interest.

The paper also concludes that ‘COVID-19 did not significantly influence the examinees’ attitude towards the exam’. On the assumption that the statistical analysis is correct, this conclusion needs further motivation. The reader is left without a clear sense on why this conclusion is unexpected or important. Although the paper briefly explains the circumstances (infections, etc.) of Japan at that time, it makes no effort in connecting this with the conclusion.

Data Set

The database contains 27949+16561+7326=51836 tweets (if my recollection is correct, there is no explicit total mentioned in the paper). Although the sample size might be sufficiently large for an experimental setup, it falls short in the context of an observational setup. In particular, most social media studies from Twitter use at least two orders of magnitude greater data sets (i.e. in the order of millions or hundreds of millions).

This in isolation is not sufficient to regard the data set dissatisfying. It could well be the case that the Twitter activity for the research question is not large. But there is no reasonable effort to justify why this data set is fitting for the research question. Some notes on this:

Why can we expect that Twitter conversation reflects the actual sentiment of students taking or about to take the exam?

What is the total number of students taking the exam, and the total number of Twitter users posting tweets? The paper does not give the number of users, and ~50K tweets can be composed by very few users, not representative of the student population.

It is not clear at all why Twitter data is better than questionnaire data, as argued in the paper. In particular, it is not clear how it avoids biases (that are present in questionnaire, or are new), as it is claimed in the paper.

Methodology

The methodology used is presented in pages 8 and 9.

I found the explanation of the methods used very obscure. Proper formulas (rather than Excel formulas) should be used in a paper. Although I am sympathetic towards Excel, programming languages like R or Python are more reliable and powerful. At the core, although some statistical knowledge can be assumed from the reader, much more clarity in explaining the methods would be required.

There are also multiple minor points on the methodology that I will list in the following section.

Minor Comments

This is simply a list of minor comments to the paper, in order of occurrence:

Abstract: The terminology “Artificial Intelligence” is too ambiguous, and the methods used for natural language processing are not state of the art (say transformer neural networks), so there is no clear sense why that is the right terminology.

The abstract is non-standardly long, and has some irrelevant information.

Pg 4. Some of the phrasing is unclear, e.g.: “student hesitation for COVID-19 pandemic has been reported.

Pg 4. I mentioned before that the case that Twitter data is more reliable than questionnaires needs to be made more clearly and strongly.

Pg 5. Some details of the sample (say the tweet items/keys connected) should go on an appendix (if any), and not in the main body.

Pg 5. Typo: ‘Titter’

Pg. 5 Unclear: “tweets were randomly collected by running the code repeatedly in Python’

Pg 6. I mentioned before that the sample size justification needs more development.

Pg 6. Did the areas where state of emergency was declared exhibited more Twitter activity?

Pg 7. An explanation of KH coder needs to be given.

Pg 7. A detailed explanation of which symbols were deleted is not necessary for the main body of the paper.

Pg 7. How were the categories for cross-tabulation determined? And why?

Pg 9. The explanation of correspondence analysis has the parameter 2 as a distance, but that is somewhat confusing with the chi-squared test mentioned before. In general, as explained in the main comment, the explanation of the methodology is not clear.

Pg 12. I suggest illustrating the correspondence pointed at the start of the paragraph in one single plot, and not part of it in Figure 1 and part in Table 2.

Pg 13 and 14. It is unclear why to include such a large table, if most of the p-values are below statistical significance.

Figures could use higher resolution for readability.

Reviewer #4: 1. Introduction

I share your passion for advancing a study on COVID-19 and its impact on the national examination for pharmacists in Japan: An SNS text analysis. A comprehensive argument is required to explain how COVID-19 impacted the national examination for pharmacists. The front end of the paper is fragmented and fails to locate a convincing gap in literature. The presentation of existing literature is fragmented and does not allow you to set up a compelling research problem.

2. Literature Review

• The study should have a concise literature review on the study variables

• A comprehensive literature support is required to justify for the selection of emotional recognition text components used in this study (anger, sadness, fear, like, and happiness) (page 15, lines 121-122). What is the uniqueness of this components?

• The study requires a well-structured hypothesis.

3. Methodology

• Author(s) should indicate the rationale for the selection of the respondents. Furthermore, the sample size technique used in determining the size of the study was not indicated. Author(s) should indicate the sample size technique employed in determining the sample size of the study.

• Author(s) indicated using Artificial Intelligence (AI) to evaluate the emotional register of the tweets on the basis of five components. The justification for the use of Artificial Intelligence (AI) should be provided.

• The techniques adopted in handling common method variance in the study was not indicated in the manuscript. Author should indicate whether the issue of method bias was significant or not significant in the study (see e.g., Conway and Lance, 2010 and Podsakoff et al.,2012).

Data Analysis

• Author(s) indicated applying an emotional recognition test. No justification for the application of the text was offered in the manuscript. Author(s) should justify the application of the text. Similar justification should be provided for the use of KH coder.

• Values of the data analysis were not reported. Author(s) should report values obtained during the analysis with appropriate literature support.

• A more robust statistical technique should be employed in testing the impact analysis by author(s).

4. Discussion and Practical Implication

The result findings are not able to justify the supporting postulations. Author(s) just offered a narrative explanation which make the discussion segment of the paper not convincing. The practical implications of the study should be carefully looked at since the main thrust of the study the COVID-19 and its impact on the national examination for pharmacists was not clearly addressed in the study.

6. PLOS authors have the option to publish the peer review history of their article (what does this mean?). If published, this will include your full peer review and any attached files.

Reviewer #1: No

Reviewer #2: **Yes: **Massimiliano Ferrara

Reviewer #3: No

Reviewer #4: **Yes: **Frank Nana Kweku Otoo

---

## [Author Response · Author response to Decision Letter 0]

1 May 2023

Thank you very much for your kind the constructive comments of the reviewer. I am most grateful for the detailed review. I have made corrections and additions according to the suggestions of the reviewers. The specific key corrections and additions are described on file.

---

## [Decision Letter · Decision Letter 1]

25 May 2023

PONE-D-22-28806R1COVID-19 and its impact on the national examination for pharmacists in Japan: An SNS text analysisPLOS ONE

Dear Dr. Kitayama,

Thank you for submitting your manuscript to PLOS ONE. After careful consideration, we feel that it has merit but does not fully meet PLOS ONE’s publication criteria as it currently stands. Therefore, we invite you to submit a revised version of the manuscript that addresses the points raised during the review process. Please submit your revised manuscript by Jul 09 2023 11:59PM. If you will need more time than this to complete your revisions, please reply to this message or contact the journal office at plosone@plos.org. Please include the following items when submitting your revised manuscript:A rebuttal letter that responds to each point raised by the academic editor and reviewer(s). You should upload this letter as a separate file labeled 'Response to Reviewers'.A marked-up copy of your manuscript that highlights changes made to the original version. You should upload this as a separate file labeled 'Revised Manuscript with Track Changes'.An unmarked version of your revised paper without tracked changes. You should upload this as a separate file labeled 'Manuscript'.If applicable, we recommend that you deposit your laboratory protocols in protocols.io to enhance the reproducibility of your results. Protocols.io assigns your protocol its own identifier (DOI) so that it can be cited independently in the future. For instructions see: https://journals.plos.org/plosone/s/submission-guidelines#loc-laboratory-protocols. Additionally, PLOS ONE offers an option for publishing peer-reviewed Lab Protocol articles, which describe protocols hosted on protocols.io. Read more information on sharing protocols at https://plos.org/protocols?utm_medium=editorial-email&utm_source=authorletters&utm_campaign=protocols.

We look forward to receiving your revised manuscript.

Kind regards,

Michal Ptaszynski, PhD

Academic Editor

PLOS ONE

Journal Requirements:

Reviewers' comments:

Reviewer's Responses to Questions

**Comments to the Author**

1. If the authors have adequately addressed your comments raised in a previous round of review and you feel that this manuscript is now acceptable for publication, you may indicate that here to bypass the “Comments to the Author” section, enter your conflict of interest statement in the “Confidential to Editor” section, and submit your "Accept" recommendation.

Reviewer #3: (No Response)

Reviewer #4: (No Response)

2. Is the manuscript technically sound, and do the data support the conclusions?

Reviewer #3: Yes

Reviewer #4: Yes

3. Has the statistical analysis been performed appropriately and rigorously? 

Reviewer #3: Yes

Reviewer #4: Yes

4. Have the authors made all data underlying the findings in their manuscript fully available?

Reviewer #3: Yes

Reviewer #4: Yes

5. Is the manuscript presented in an intelligible fashion and written in standard English?

Reviewer #3: Yes

Reviewer #4: Yes

6. Review Comments to the Author

Reviewer #3: Second Review

‘COVID-19 and its impact on the national examination for pharmacists in Japan: An SNS text analysis’

Brief summary of the paper

In this work the authors used a data set obtained from Twitter in order to assess the effect that COVID-19 had on the national examination for pharmacists in Japan. Using natural language processing techniques, in particular sentiment analysis, they concluded: “In the first half of December, COIVD-19 -related words were detected in association with fear emotions, but since then mainly exam-related words have been associated with negative emotions. On the other hand, examinees in high infection areas that had a state of emergency were found to be anxious about their connection with society, including "employment". This trend was less pronounced in other areas”

Recommendation

The paper is much improved from the first version. Provided some changes are made to it, I do think it is suitable for publication.

Side Comments

As a general practice, it would for the author to submit a response letter addressing each of my previous comments in particular. If they did, I do not think I received it.

Comments

As I mentioned, the paper greatly improved in writing clarity and in addressing the limitations that I stated in my first review. In particular, the results are explained much better and the visualizations are compelling (although some color would be appreciated).

The main concern that I have is not with the main body of the text, but rather with the motivation in the introduction and the conclusions at the end.

On the one hand, the study could very much profit from a better motivation. What are the research questions guiding the investigation? Why are the observations revealed by the analysis important or useful? To be clear, I think that the contribution is valuable, but its motivation is not clearly explained. For example, a natural motivation would be to argue that it is important to study how the medical community, and in particular future practitioners, reacted to a global pandemic, so as to make sure things are improved in an eventual next time. Where students deterred or invited to pursue their careers? Etc.

On the other hand, the conclusion is very interesting but narrow and short. This is related with the point about the motivation. As a reader I was left wanting for a more developed account of the points made in the conclusion. For example, I found this very interesting: “The results of this study indicate that due to the impact of the COVID-19 pandemic, attention should be paid to social relevance, including the future of examinees, rather than the examination itself. Such anxieties are likely to make students feel embarrassed and hesitant about working as medical professionals because of COVID-19.” This is interesting because it suggests that the path to a better situation is not focusing so much on covid, but on the student’s emotions. Is there literature suggesting more in this direction? More generally, it would be nice to see a general recommendation about how to respond to these scenarios that builds both on present literature and the particular observations from the analysis presented.

Some minor comments are the following:

- Some references connecting with the effect of covid on other student populations would be useful.

- I am confused why are doctors mentioned if the paper focuses on pharmacists.

am not sure the contrast with questionnaires helps your case here. Also, maybe more references would be necessary.

- ‘Mechanically’ (line 66) does not seem the right way. Maybe ‘automatically’?

- Lines 95-115. Too much detail in the description. It is good to mention the particular libraries used, but possibly not in the main text as it makes it more convoluted and harder to read.

- Formulas around line 132: This is much better than before. But what are the i,j? What are a and b?

- Limitations Section: Do you know if those who posted where those getting examined? If so, how? Still, this is a very important point. The proportion of users is sufficiently large.

Reviewer #4: The manuscript has seen a thorough and in-depth revision to justify publication in its current form

7. PLOS authors have the option to publish the peer review history of their article (what does this mean?). If published, this will include your full peer review and any attached files.

Reviewer #3: **Yes: **Ignacio Ojea Quintana

Reviewer #4: **Yes: **frank nana kweku otoo

---

## [Author Response · Author response to Decision Letter 1]

8 Jun 2023

Thank you very much for your constructive comments of the reviewer. I am most grateful for the detailed review. I have gone over the comments carefully, and made corrections and additions according to the suggestions of the reviewer.

---

## [Decision Letter · Decision Letter 2]

19 Jun 2023

COVID-19 and its impact on the national examination for pharmacists in Japan: An SNS text analysis

PONE-D-22-28806R2

Dear Dr. Kitayama,

We’re pleased to inform you that your manuscript has been judged scientifically suitable for publication and will be formally accepted for publication once it meets all outstanding technical requirements.

Kind regards,

Michal Ptaszynski, PhD

Academic Editor

PLOS ONE

Additional Editor Comments (optional):

Reviewers' comments:

Reviewer's Responses to Questions

**Comments to the Author**

1. If the authors have adequately addressed your comments raised in a previous round of review and you feel that this manuscript is now acceptable for publication, you may indicate that here to bypass the “Comments to the Author” section, enter your conflict of interest statement in the “Confidential to Editor” section, and submit your "Accept" recommendation.

Reviewer #2: All comments have been addressed

2. Is the manuscript technically sound, and do the data support the conclusions?

Reviewer #2: Yes

3. Has the statistical analysis been performed appropriately and rigorously? 

Reviewer #2: Yes

4. Have the authors made all data underlying the findings in their manuscript fully available?

Reviewer #2: Yes

5. Is the manuscript presented in an intelligible fashion and written in standard English?

Reviewer #2: Yes

6. Review Comments to the Author

Reviewer #2: The paper after revision can be accepted for publication. The Authors have addressed successfully all requests as well

7. PLOS authors have the option to publish the peer review history of their article (what does this mean?). If published, this will include your full peer review and any attached files.

Reviewer #2: **Yes: **Massimiliano Ferrara

---

## [Editor Report · Acceptance letter]

23 Jun 2023

PONE-D-22-28806R2 

COVID-19 and its impact on the national examination for pharmacists in Japan: An SNS text analysis 

Dear Dr. Kitayama:

I'm pleased to inform you that your manuscript has been deemed suitable for publication in PLOS ONE. Congratulations! Your manuscript is now with our production department. 

Kind regards, 

on behalf of

Dr. Michal Ptaszynski 

Academic Editor

PLOS ONE